# A New Setup for the Measurement of Total Organic Carbon in Ultrapure Water Systems

**DOI:** 10.3390/s22052004

**Published:** 2022-03-04

**Authors:** Sara H. Schäfer, Katharina van Dyk, Johannes Warmer, Torsten C. Schmidt, Peter Kaul

**Affiliations:** 1Department of Natural Science, Safety and Security Research Institute, Bonn-Rhein-Sieg University of Applied Science, 53359 Rheinbach, Germany; peter.kaul@h-brs.de; 2Innovatec Gerätetechnik GmbH, 53359 Rheinbach, Germany; k.vdyk@innovatec-rheinbach.de (K.v.D.); j.warmer@innovatec-rheinbach.de (J.W.); 3Department of Instrumental Analytical Chemistry, University of Duisburg-Essen, 45147 Essen, Germany; torsten.schmidt@uni-due.de

**Keywords:** TOC, ultrapure water, ozonation, AOP, O_3_/UV

## Abstract

With the increasing demand for ultrapure water in the pharmaceutical and semiconductor industry, the need for precise measuring instruments for those applications is also growing. One critical parameter of water quality is the amount of total organic carbon (TOC). This work presents a system that uses the advantage of the increased oxidation power achieved with UV/O_3_ advanced oxidation process (AOP) for TOC measurement in combination with a significant miniaturization compared to the state of the art. The miniaturization is achieved by using polymer-electrolyte membrane (PEM) electrolysis cells for ozone generation in combination with UV-LEDs for irradiation of the measuring solution, as both components are significantly smaller than standard equipment. Conductivity measurement after oxidation is the measuring principle and measurements were carried out in the TOC range between 10 and 1000 ppb TOC. The suitability of the system for TOC measurement is demonstrated using the oxidation by ozonation combined with UV irradiation of defined concentrations of isopropyl alcohol (IPA).

## 1. Introduction

The quality of ultrapure water is of particular importance in the pharmaceutical and semiconductor industry. Due to its unique properties, it serves as a solvent or starting product for the preparation of injection solutions, and it is also used to clean surfaces in the production of high-precision components in semiconductor manufacturing. Contamination of any kind leads to considerable loss in the quality of the manufactured products in semiconductor industry, and impurities in water for injections would have a direct impact on human health [1,2]. For this reason, water quality is of utmost importance and requires monitoring.

An important parameter for the quality of ultrapure water is the total organic carbon (TOC). TOC is defined as the carbon content of dissolved (DOC) and non-dissolved (NDOC) organic carbon present in water. DOC is furthermore defined as the sum of organically bound carbon after membrane filtration with a filter of 0.45 µm pore size [3]. Because water is filtered in an ultrapure water system, the influence of the NDOC is irrelevant for this work and, consequently, no further distinction is made here between TOC and DOC. TOC is a sum parameter that serves as an indicator for the extent of organic pollution in water. Organic substances are the most common contaminants in water [4], thus the TOC is a key criterion for the water quality of ultrapure water and the measurement of TOC allows an estimation of the degree of pollution of water in general. As ultrapure water has only trace amounts of organic pollution, a very sensitive method is needed for water analysis. Thresholds and standards for the analysis of TOC are defined by the European Pharmacopeia [5], the United States Pharmacopeia (USP) [6], the Deutsches Institut für Normung [3] and the American Society for Testing and Materials [7,8,9,10,11,12,13].

For water used in the pharmaceutical industry, not only the TOC content, but also the temporal development of the TOC due to biofilm generation is a parameter of interest concerning the quality control [4].

Conventional methods for TOC measurements are based on thermal combustion of carbonaceous substances or on UV oxidation (catalyzed and not-catalyzed) [14,15]. All common methods are based on the total oxidation of the present organic carbon (OC) and the subsequent detection of the resulting CO_2_ generation. In pharmaceutical water, where the addition of chemicals has to be avoided, an approach for the determination of lower TOC loads utilizes oxidation by UV-C irradiation [16]. In comparable measuring instruments, for instance, the water is passed around a high-energy UV source for oxidation in a quartz glass spiral and irradiated with UV light of wavelength 185 nm [17]. The irradiation with such high-energy UV light can, on the one hand, lead to the direct photolysis of the target substances. On the other hand, the irradiation of water with this wavelength leads to the photolysis of H_2_O resulting in the generation of hydroxyl radicals (^●^OH), which subsequently oxidize organic compounds (see Equation (1)) [18,19].
(1)2 H2O →hv H2+2 O●H

The novel approach, described in this work, lies in the combination of oxidation with UV irradiation and ozonation for the measurement of TOC. The UV/O_3_ process is referred to as an AOP [20,21], and it is a well-established method for degradation of organic compounds in water treatment [22]. Thus, the advantage of several possible oxidation pathways can be used in the new measuring system since the strong oxidant O_3_ is additionally available for oxidation. O_3_ has an absorption maximum at a wavelength of 260 nm. The exposure of UV light of this wavelength to O_3_ in water causes O_3_ to decompose and form ^●^OH, as shown in the following Equations (2) and (3) [23,24]. In the first step, O_3_ decomposes into molecular oxygen and oxygen atoms in the two major spin-allowed reactions, shown in Equations (2) and (3).
(2)O3 →hv260 O(D1)+O2(Δ1g)
(3)O3 →hv260 O(P3)+O2(∑g−3)

The resulting species differ in O(^1^D) (oxygen atom in excited state), O(^3^P) (oxygen atom in ground state), singlet oxygen O_2_(^1^Δ*_g_*), and ground state oxygen O_2_(^3^${\sum^{}}_{g}^{-}$).

O(^1^D) is highly energetic and reacts in a second step with water to form thermally excited hydrogen peroxide (H_2_O_2_) “hot”, which furthermore thermalizes to H_2_O_2_ or decomposes to form two ^●^OH (see Equations (4)–(6)).
(4)O(D1)+H2O→ H2O2 (hot)
(5)$$H_{2}O_{2}\;\left(hot\right)\;\to \;H_{2}O_{2}$$
(6)H2O2 (hot) → 2O●H

The formed H_2_O_2_ also contributes to the formation of ^●^OH. On the one hand, it can react with O_3_ in the so-called peroxone process, resulting in the formation of ^●^OH. On the other hand, it can undergo photolysis and thus form radicals [23].

The overall ^●^OH quantum yield of the UV/O_3_ reaction is 0.1 [21,23,24]. According to von Sonntag et al. [24] and Fischbacher et al. [23], the low quantum yield of this reaction is attributed to recombination reactions in the solvent cage (Equations (7) and (8)).
(7)O(D1)+O2(Δ1g) → O3
(8)[ O●H+O●H ] → H2O2

The UV/O_3_ system offers the possibility of different ways of oxidation in the newly developed measuring instrument. As in the direct UV oxidation, the existing organic carbon (OC) can undergo a photolysis. It can furthermore be oxidized directly by O_3_ or by ^●^OH. However, the dominant oxidation process begins with the splitting of O_3_ into ^●^OH initiated by UV radiation, as ^●^OH reacts much faster with IPA than O_3_. Reisz et al. report that the reaction rate of IPA with O_3_ (k (IPA + O_3_) = 2.7 M^−1^ s^−1^) is roughly by a factor of 10^9^ smaller than the reaction rate of IPA with ^●^OH (k (IPA + ^●^OH) = 1.2 × 10^9^ M^−1^ s^−1^) [25].

Based on the described limitations of existing systems, the aim of this work was to perform a proof-of-concept for the development of a TOC measuring instrument that enables the online measurement of low-level TOC in ultrapure water with a limit of detection (LOD) of 50 ppb, as required by the USP [6]. In the presented setup, the TOC was correlated with conductivity changes before and after oxidation of OC. The recent development in LED technology makes the use of UV-LEDs instead of mercury vapor lamps possible. This leads to significant savings in energy and material costs and also contributes to the miniaturization of the setup.

## 2. Materials and Methods

The core of the setup as shown in Figure 1 is a 200 mm long quartz glass tube with an inner diameter of 4 mm and a wall thickness of 1 mm. This tube serves as the reaction chamber and irradiation section and is positioned between two blocks made from PTFE. O_3_ is generated in situ in water with a so–called “ozone microcell (OMC)” developed by Innovatec Gerätetechnik GmbH [26]. This PEM-O_3_ generator is integrated in the lower PTFE block and generates O_3_ electrolytically directly at the point-of-use. The amount of produced O_3_ is current-controlled with a maximum production of 15 mg O_3_ per hour at 200 mA electrolysis current [26]. 

The water used is taken from an ultrapure water system, consisting of a water softener (BM16, AQMOS, Seligenstadt, Germany) an electrodeionization (SEPTRON SM 10, BWT, Bietighei-Bissingen, Germany), a reverse osmosis (Osmotech Ultimate Plus Superflow, Filterzentrale, Hohenbrunn, Germany) and a UV lamp (WEDECO Aquada, Xylem, Langenhagen, Germany). The water passes the ozone generator and enters the quartz glass tube. For irradiation, 5 surface-mounted device LEDs (CUD8AF4D—Seoul Viosys, Neumüller, Weisendorf, Germany) with a wavelength of 275 nm are used. These LEDs are used as they represent, in accordance with the current state of the art, a good compromise between the achieved light power and the absorption coefficient of ozone. The LEDs are attached to one side of the quartz glass tube. The water is ozonated by the OMC and directed into the glass tube. After passing the irradiation section, formed gas bubbles are separated from the water by a PTFE membrane. Thereafter, the conductivity and the water temperature are measured. For the temperature measurement, a Pt1000 sensor (Kabelfühler—3 mm, Sensorshop24, Bräunlingen, Germany) is used. The conductivity is measured with self-designed electronics. The design of the measuring cell shown in Figure 2.

The measuring cell is a flow cell, which is made of PTFE. It has G 1/8 “ threads on both sides, so that fittings can be connected. The stainless-steel electrodes are pressed into the measuring cell from both sides. They are 20 mm long and have a distance of 0.2 mm from each other. Measurements are taken with a rectangular voltage of 168 Hz with an amplitude of ±1 V. The measuring principle of the system is the evaluation of the changes in conductivity. For this purpose, the conductivity is measured continuously and the difference between the unoxidized sample and the sample after oxidation is evaluated. The assumption is that the total difference in conductivity results from the oxidation of the dissolved OC. A defined quantity of OC is continuously added to the water via a syringe pump. Afterwards, it is mixed to guarantee a homogenous solution (see the mixing unit in Figure 1). The flow rate through the system is constant at 10 mL/min. The variation of the volume flow of the syringe pump or of the concentration of the added stock solution leads to a defined change in OC concentration. After changing the settings, a minimum of 180 min is selected between two measurements with different OC concentrations in order to achieve a new constant concentration.

For all experiments, IPA (99.8%, Merck, Darmstadt, Germany) serves as organic carbon model substance as it is widely used in semiconductor manufacturing during drying processes, and it is oxidized preferably via ^●^OH-oxidation [27]. Wu et al. [28] showed the successful degradation of IPA with UV/O_3_ oxidation and Reisz et al. [25] evaluated the possible pathways of O_3_ and ^●^OH attack on IPA.

One measurement cycle consists of five irradiation periods (each 120 s) followed by periods without irradiation (180 s). The conductivity and the temperature of the water are measured permanently. Two conductivity and temperature measuring points are recorded per second and the conductivity is temperature compensated with an increase of 2% of conductivity per 1 °C of temperature [29]. The ozonated solution is further treated by applying a total current of 400 mA to the five UV-LEDs, which corresponds to an optical power of about 32 mW.

OriginLab Origin^®^ 2021b is used for data evaluation and presentation.

## 3. Results and Discussion

The reproducibility of the measurement at a constant IPA concentration is shown in Figure 3, which illustrates the conductivity changes over time during one complete measuring cycle of the oxidation of a solution with 19.5 ppb_C_ IPA.

Starting from the baseline, the irradiation of the ozonated solution results in an increase in conductivity of the solution during irradiation periods. The irradiation of the ozonated water leads to the formation of highly reactive ^●^OH, which subsequently reacts with IPA to form acidic oxidation products [28,30]. The rapid increase can be explained by the considerable differences in the reaction rates of IPA with O_3_ and ^●^OH [25].

Stefan et al. [31], Wu et al. [28] and Choi et al. [32] identified acetone as the first step degradation product in the degradation pathway of IPA via O_3_/UV AOP. Acetic acid, oxalic acid and formic acid are identified as the secondary degradation intermediates, which are subsequently oxidized to CO_2_. Reisz et al. [25] additionally identified acetaldehyde, formaldehyde and hydrogen peroxide in the degradation pathway. Both the acidic intermediates and CO_2_ dissolved in water, which lead to an increase in conductivity of the analyzed water.

Concentration-dependent measurements were used to correlate TOC with conductivity change. To evaluate the influence of the TOC concentration on the conductivity profile, the first 100 s of irradiation are considered. Figure 4 shows the conductivity profiles of four differently concentrated solutions.

The mean increase in conductivity of the five successive measurements per concentration is plotted with the standard deviations over time for the measurements of four differently concentrated solutions. For better comparability between the different measurements, the measured conductivity was offset corrected for each irradiation cycle by subtracting the first value of the measuring cycle from the following ones. It is clearly apparent from Figure 4 that the conductivity increases with increasing added organic carbon and can be well distinguished from each other after 100 s of irradiation. Since the aim is to record the TOC content of water in an online mode, an evaluation in shorter time intervals would be more appropriate. For this reason, in addition to the conductivity-changes after 100 s of irradiation Δσ (see Figure 3), the initial slopes were also evaluated. This also has the advantage that not only is a single point used for evaluation, but a fit to the curve is made, which makes the evaluation less error-prone. The results of this evaluation are shown in Figure 5.

Here, trend lines are fitted linearly to the first 20 s of measurement of the averaged increase in conductivity of the five successive measurements, again for the four differently concentrated solutions. The slopes derived from the trend lines are given in Table 1, with the respective correlation coefficient.

The slopes clearly increase with increasing initial OC concentration, from 1.13 × 10^−3^ for the solution without addition of OC, to 3.26 × 10^−3^ for the solution with 225 ppb, and the correlation coefficient is above 99% for all four linear fits. This shows that a differentiation based on the difference after 100 s of irradiation and via the slope evaluation in the initial range of the irradiation is well feasible It has to be mentioned that in the solution without OC addition, an increase in the conductivity can be observed (see Figure 4 black curve). However, both the height of the increase and the initial slope during irradiation are lower than the values of the solution to which 10 ppb OC were added. The label “0 ppb” means only that no OC was added. Residual contamination cannot be avoided, so this increase can be attributed to a residual, but unknown, amount of impurities in the water.

Additionally, the effect of UV irradiation alone on the increase in conductivity was tested by comparing a solution with 50 ppb added OC, which was irradiated, with and without ozonation. The results are plotted in Figure 6.

The comparison of the mean increase in the five successive measurements of the solution containing 50 ppb_C_ IPA with (black) and without (red) ozonation shows only a very small increase of approx. 2.5 × 10^−3^ µS/cm during UV irradiation without ozonation of the solution. The solution that was oxidized with UV/O_3_, shows a significantly higher increase in conductivity (approx. 0.10 µS/cm). The significant increase in conductivity after irradiation of the ozonated solution shows that the organic compounds are degraded and acidic oxidation products are formed, which are detected by the conductivity measurement. This proves a successful degradation of IPA via ^●^OH oxidation and shows that the UV oxidation pathway similar to direct ozonation is negligible for this substance, as expected.

The increase in conductivity after 100 s of irradiation served for the correlation with the OC concentration. This value was determined for differently concentrated IPA solutions in the range between 10 and 750 ppbC.

The mean values of the increase in the baseline corrected conductivity after 100 s of oxidation are plotted in Figure 7 as a function of the respective added OC. The five single measurements for each data point after 100 s were controlled in a confidence interval of 99% for outliers according to Dixon [18]. The dataset is divided into 20 training data, used for calibration, and 3 test data sets. The training data set was used for correlation between Δσ and added OC and for the calculation of a 99% confidence and prediction band.

As a result, an almost logarithmic correlation was observed between the IPA concentration and the increase in conductivity after 100 of irradiation. This correlation can be further used for the determination of TOC in ultrapure water.

Using the three test data points, the calculated correlation between added OC and increase in conductivity was verified. For this purpose, test measurements of solutions with an added OC of 10 ppb, 25 ppb and 40 ppb were conducted and the mean increase after 100 s of irradiation is plotted with green data points in Figure 7. Since the USP [31] requires a LOD of 50 ppb, testing of the system is performed in the lower concentration range. There is no significant difference between the calculated and measured points. All three measured points are within the 99% prediction band, which shows that it is possible to use the newly developed system to successfully correlate the change in conductivity during oxidation with the TOC between 10 and 1000 ppb.

The same evaluation was made with the slopes in the initial range of the measurements. For this purpose, the slopes after 5 s, 10 s, 15 s and 20 s of irradiation were evaluated and plotted in Figure 8.

Based on the correlation coefficients from the slope evaluations, it can be seen that linear fitting to the slope after 5 s only yields a correlation of 53%. With increasing length of evaluation interval, the correlation coefficient also increases. Thus, the evaluation of the slope of the first 20 s of the measurement already provides a correlation coefficient of 85%. A comparison of the slopes shows that the slopes in the evaluations after 5 s, 10 s and 15 s differ from each other. A further extension of the evaluation interval by 5 s to a total of 20 s does not lead to any significant change. Thus, the evaluation of the slope after 15 s provides a good compromise between the length of the evaluation interval and the correlation coefficient.

## 4. Conclusions

In summary, the newly developed system offers three main advantages: (i) it does not require the addition of chemicals—ozone, which is used for oxidation, is generated in situ, and decomposes without leaving residues; (ii) using UV LEDs allows the new system, with an overall height of 25 cm, to require a much smaller footprint than, for example, the TOC measuring device from SWAN [33]; the latter is 40 cm wide and 85 cm high and is thus considerably larger; therefore, the new system can be used in different locations without any problems; (iii) it enables online monitoring of low-level TOC. At present, differentiation based on the differences after 100 s of irradiation provides the most accurate evaluation with a correlation coefficient of almost 95%.

It was demonstrated, for further development and optimization of the measuring instrument, that differentiation based on the slopes is also conceivable.

Furthermore, the high oxidation power and the additional degradation pathways allow the oxidation of a wide range of analytes.

Altogether, it was shown for IPA that with the help of the newly developed measuring system it is possible to measure TOC in the range between 10 and 1000 ppb. Further validation measurements must be taken to verify the system’s suitability for the detection of different substance classes.

## 5. Patents

The patent with the number DE 102019107260 A1 resulted from the work reported in this manuscript.

## Figures and Tables

**Figure 1 sensors-22-02004-f001:**
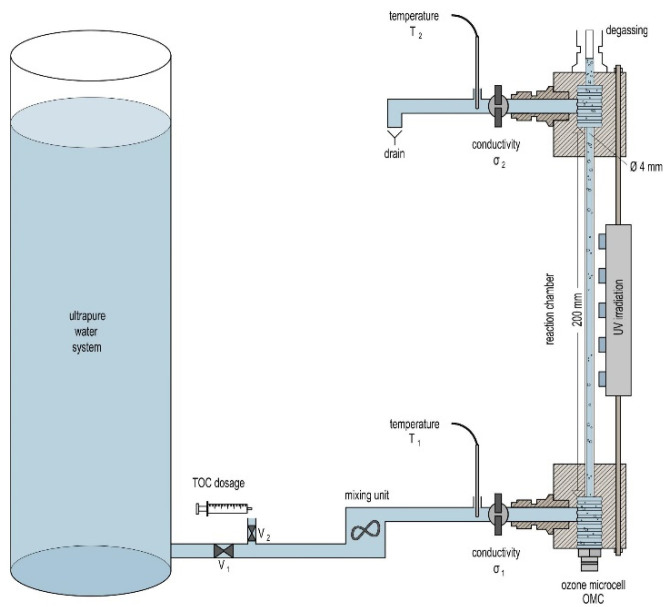
Schematic representation of the overall setup for TOC measurement. For a better overview, the actual measuring cell is shown enlarged in comparison to the ultrapure water system. There is no size proportionality. The measuring block with conductivity and temperature measurement is approx. 25 cm in size.

**Figure 2 sensors-22-02004-f002:**
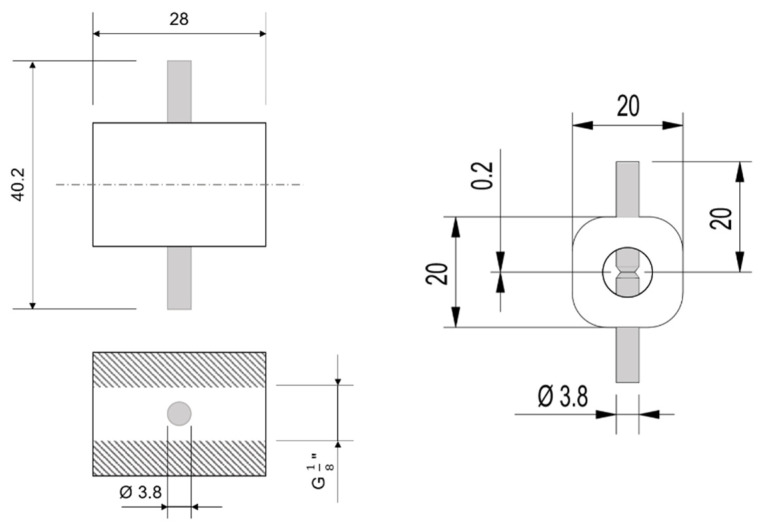
Construction drawing of the measuring cell used for conductivity measurement. Side view on the left and top view on the right side. The measuring cell is made of PTFE (white parts), the electrodes of stainless steel (grey parts). The lengths are given in mm.

**Figure 3 sensors-22-02004-f003:**
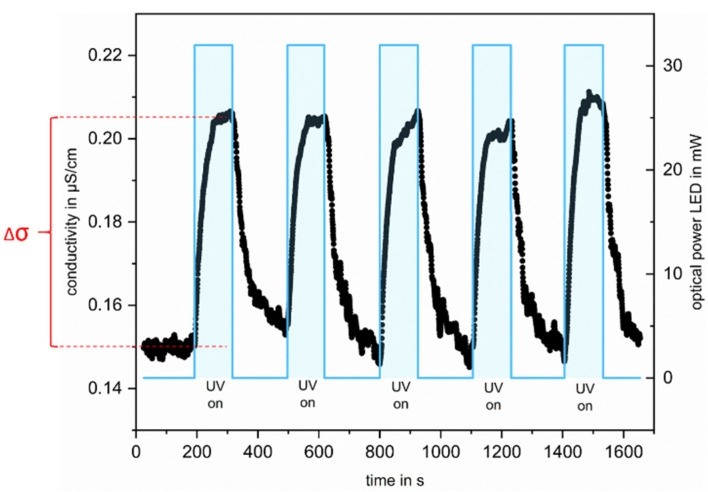
Measured conductivity (black lines) of a solution containing 19.5 ppb TOC (source of carbon: IPA) during five successive UV/O_3_ oxidation cycles. The solution was constantly ozonated with an operating current of 200 mA. In the irradiation periods, the solution was irradiated with an optical power of 32 mW. In each repetition, the LEDs were switched off for three minutes, followed by two minutes of UV irradiation (marked with the blue bars). The increase in conductivity during irradiation is marked in red as Δσ.

**Figure 4 sensors-22-02004-f004:**
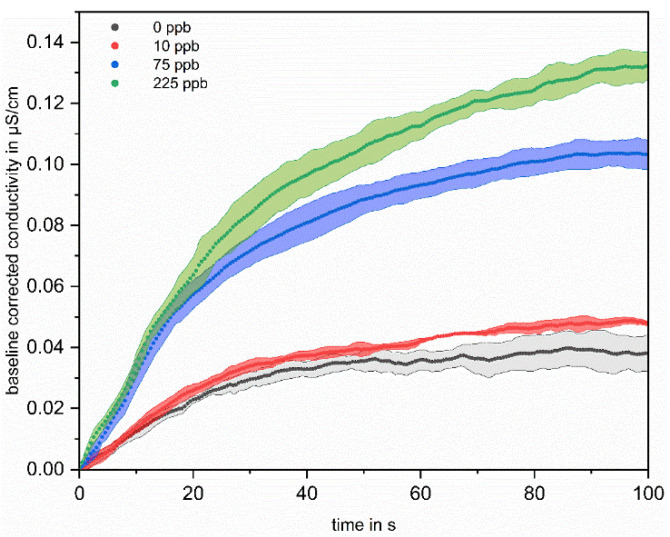
Mean values and standard deviations of baseline corrected conductivity measurements during first 100 s of irradiation. The mean values and standard deviations refer to the five measurements per concentration level. The black curve shows the course of the solution without added OC, the red curve shows the conductivity for 10 ppb, the blue curve for 75 ppb and the green curve for 225 ppb added OC. For offset correction, the first value of the measuring cycle was subtracted from the following values to obtain the difference in conductivity.

**Figure 5 sensors-22-02004-f005:**
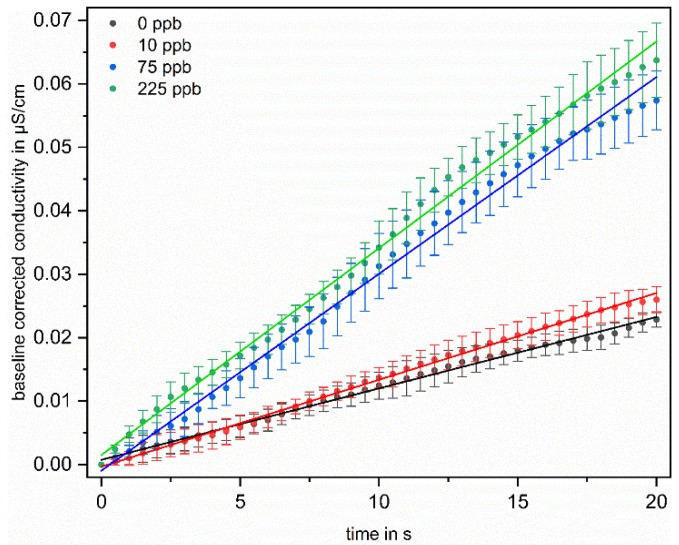
Mean values and standard deviations of baseline corrected conductivity measurements during first 20 s of irradiation with linear fitted trend line. The mean values refer to the five measurements per concentration level. The black curve shows the course of the solution without added OC, the red curve shows the conductivity for 10 ppb, the blue curve for 75 ppb and the green curve for 225 ppb added OC. For offset correction the first value of the measuring cycle was subtracted from the following values to obtain the difference in conductivity.

**Figure 6 sensors-22-02004-f006:**
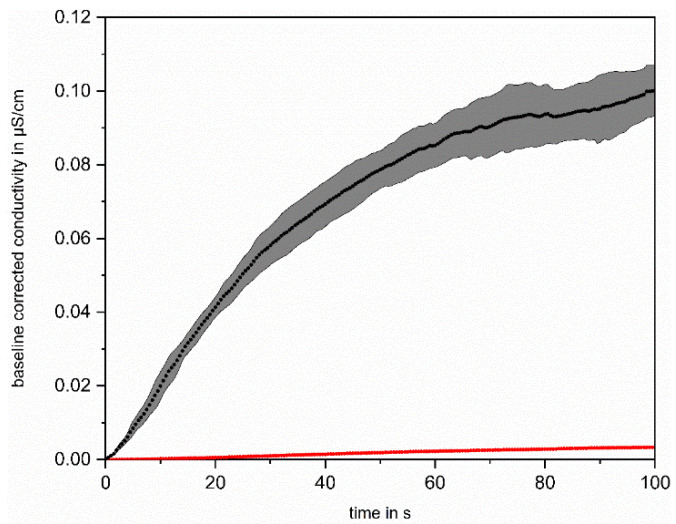
Comparison of the mean increase in conductivity during UV irradiation of a solution with 50 ppb_C_ IPA without ozonation (red) and with ozonation (black).

**Figure 7 sensors-22-02004-f007:**
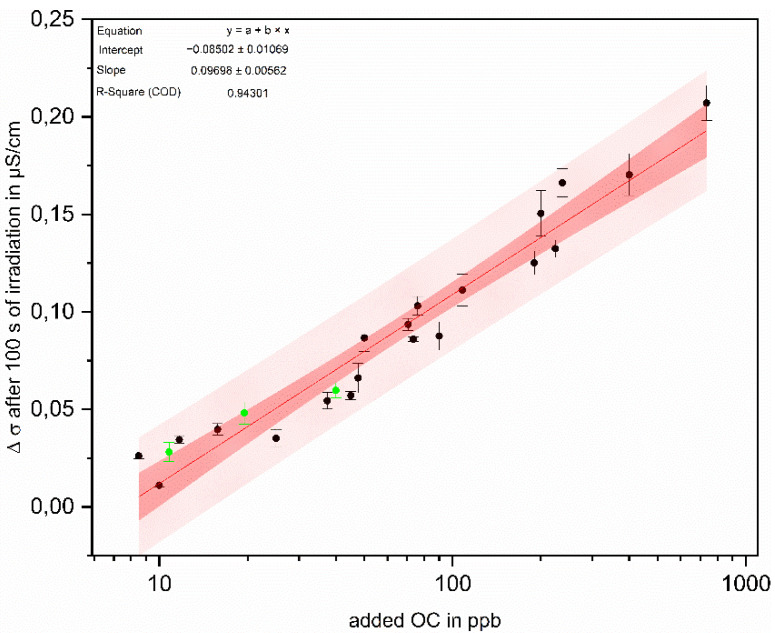
Logarithmic correlation of added OC and baseline corrected conductivity after 100 s of irradiation for 20 different training data (black dots, *n* = 5) and verification via 3 different test data (green dots, *n* = 5) with a 99% confidence (dark red area) and prediction band (light red area).

**Figure 8 sensors-22-02004-f008:**
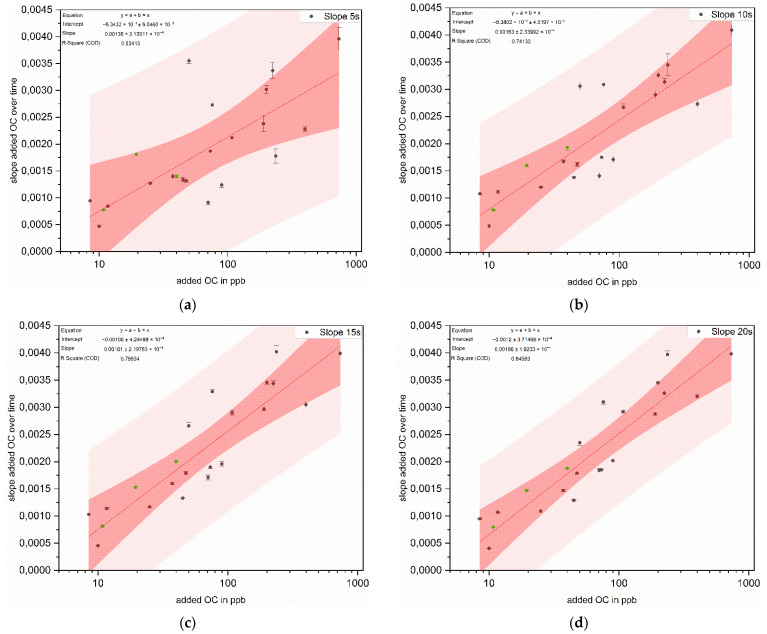
Logarithmic correlation of added OC and the slopes after 5 s (**a**), 10 s (**b**), 15 s (**c**) and 20 s (**d**) of irradiation for 20 different training data (black dots, *n* = 5) and verification via test data (green dots, *n* = 5) with a 99% confidence (dark red area) and prediction band (light red area) and the corresponding correlation coefficients.

**Table 1 sensors-22-02004-t001:** Summary of the slopes with corresponding correlation coefficient derived by linear fitting of trend lines to the first 20 s of irradiation for the measurements of four differently concentrated solutions (0 ppb, 10 ppb, 75 ppb and 225 ppb).

Added OC in ppb	Slope	R-Square in %
0	1.13 × 10^−3^ ± 1 × 10^−5^	99.55
10	1.37 × 10^−3^ ± 1 × 10^−5^	99.79
75	3.10 × 10^−3^ ± 4 × 10^−5^	99.40
225	3.26 × 10^−3^ ± 4 × 10^−5^	99.42

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
