# Peer review of "A New Setup for the Measurement of Total Organic Carbon in Ultrapure Water Systems"

_sensors, 2022, doi:10.3390/s22052004_

Round 1

Reviewer 1 Report

In this work, the authors developed a new system for TOC measurement. It combines UV-C and ozone to decompose the organic substance and measures the conductivity increase to determine the TOC. UV-LED and in-situ ozone generator help reduce the footprint of instrument and allow continuous measurement. IPA was used as a model compound to demonstrate its suitability. Overall, this is a well-designed set-up. But I think the manuscript requires major revisions to clarify the significance of this work and convince readers why it is innovative and explain what problems it can solve. In addition, some conclusions are not well supported by experimental evidence. Statistical analysis is insufficient to validate the new measurement methods. Here are my detailed comments.

  1. Why is the set-up described as “sensor-based” in the title? As far as I know, most TOC measurements are based on some kinds of sensors such as IR and conductivity. The title is misleading and needs further explanation.
  2. Please add the full name of PEM in the abstract.
  3. In the introduction, the authors should briefly talk about why TOC is an important parameter and how it affects the semiconductor and pharmaceutical industry.
  4. The authors should provide the background about the current TOC measurement techniques. So, we would know what limitations the existing instruments have and what problems the present system is trying to solve. The authors mentioned a little bit about conventional oxidation in the paper but not clear to me.
  5. Since the UV/ozone technique has been widely used for water treatment, are there any examples in the literature about their application on TOC measurement?
  6. The difference of the slopes between 75 and 225 ppb is very small. More statistical analysis is needed to support that the TOC determined by either after 100s or slope methods is reliable and reproducible. I would suggest the authors collect more data points and evaluate their accuracy and precision.
  7. In the conclusion, the authors claim that there is no ozone residue, but no evidence is given.
  8. What is the footprint of a normal TOC instrument? How does it compare with this set-up?
  9. There is still no experiment can prove it can be applied to a wide range of analytes as the authors claim.

Author Response

Dear Reviewer, 

I have summarized the responses to the comments in the attached Word document.

Best regards
Sara Schäfer

Reviewer 2 Report

The authors present a setup to detect low levels of total organic carbon (TOC) through an advanced oxidation process (AOP), a common method that involves the use of UV and ozone to generate hydroxy radicals to oxidize organics. TOC analyzers have been demonstrated commercially and in the literature to operate via UV oxidation alone or through the UV/ozone oxidation process (AOP) leveraged by the authors with very low limits of detection and portable operation (https://doi.org/10.1039/C9EW00653B, TOC Portable 450TOC 110/240VAC, StarTOC - Ozone Promoted/Hydroxyl Radical Analyzer, etc.). The authors outline their setup which involves water to be directed past an ozone generator, a tube where the water is then irradiated by UV, a membrane for degassing, and an outlet to measure conductivity (to correlate with TOC). The authors outline the purpose of each component of their setup. 

Conductivity was shown to increase with the addition of IPA to their setup as expected. Different TOC concentrations were able to be discriminated after 100 s. The use of ozonation was shown to be critical to attaining significant increases in conductivity. A calibration curve was then developed to correlate conductivity and added OC at very low concentrations (10-1000 ppb). 

Please address:

-All acronyms must be spelled out in the body, even if present in the abstract or seems obvious (AOP, IPA, LOD, USP)

-Figure 2: Please point out what material each component is made of and include a side view. The drawing as given does not enable reproducibility. 

-If it takes 100s for the conductivity to level out, does this mean that ~1-2 mL is required to get an accurate reading? Is that a lot or very little sample for the type of application this is intended for? Please comment within the manuscript.

Author Response

(The authors gave the same response as above.)

Reviewer 3 Report

Schafer et al. developed a total organic carbon (TOC) sensor based on UV/O3 oxidation and conductivity measurement. It can potentially be developed into a miniaturized and online TOC analyzer. The principle and design are sound. Therefore, I support its publication in Sensors after the following concerns are addressed.

Specific comments:

  • Only IPA was used to simulate TOC in water. The authors should discuss the problems when the sensors are applied for waters containing a complex mixture of natural organic matter.
  • Have the training and testing data set been validated using a commercial TOC analyzer?

Author Response

(The authors gave the same response as above.)

Round 2

Reviewer 1 Report

The authors addressed my comments in the first round well. I would just suggest providing the average dimension of a commercially available TOC instrument to highlight the smaller footprint of the newly developed system.

Author Response

Dear Reviewer,

I have compared the dimensions of a commercially available TOC instrument with those of the newly developed system to highlight the smaller footprint of the latter one. 

Yours sincerely
Sara Schäfer